# School-Based Nutrition Interventions in Children Aged 6 to 18 Years: An Umbrella Review of Systematic Reviews

**DOI:** 10.3390/nu13114113

**Published:** 2021-11-17

**Authors:** Kate M. O’Brien, Courtney Barnes, Serene Yoong, Elizabeth Campbell, Rebecca Wyse, Tessa Delaney, Alison Brown, Fiona Stacey, Lynda Davies, Sasha Lorien, Rebecca K. Hodder

**Affiliations:** 1School of Medicine and Public Health, University of Newcastle, University Drive, Callaghan, Newcastle, NSW 2308, Australia; courtney.barnes@health.nsw.gov.au (C.B.); serene.yoong@health.nsw.gov.au (S.Y.); libby.campbell@health.nsw.gov.au (E.C.); rebecca.wyse@health.nsw.gov.au (R.W.); tessa.delaney@health.nsw.gov.au (T.D.); alison.brown7@health.nsw.gov.au (A.B.); sasha.lorien@health.nsw.gov.au (S.L.); rebecca.hodder@health.nsw.gov.au (R.K.H.); 2Priority Research Centre in Health and Behaviour, University of Newcastle, University Drive, Callaghan, Newcastle, NSW 2308, Australia; fiona.stacey@health.nsw.gov.au; 3Hunter New England Population Health, Longworth Avenue Wallsend, Newcastle, NSW 2287, Australia; lynda.davies@health.nsw.gov.au; 4Hunter Medical Research Institute, Kookaburra Circuit, New Lambton Heights, Newcastle, NSW 2305, Australia; 5National Centre of Implementation Science, University of Newcastle, University Drive, Callaghan, Newcastle, NSW 2308, Australia; 6Faculty of Health, Arts and Design, Swinburne University of Technology, John Street, Hawthorn, VIC 3122, Australia

**Keywords:** umbrella review, school-based, intervention, dietary intake, nutrition, child, adolescent

## Abstract

Schools are identified as a key setting to influence children’s and adolescents’ healthy eating. This umbrella review synthesised evidence from systematic reviews of school-based nutrition interventions designed to improve dietary intake outcomes in children aged 6 to 18 years. We undertook a systematic search of six electronic databases and grey literature to identify relevant reviews of randomized controlled trials. The review findings were categorised for synthesis by intervention type according to the World Health Organisation Health Promoting Schools (HPS) framework domains: nutrition education; food environment; all three HPS framework domains; or other (not aligned to HPS framework domain). Thirteen systematic reviews were included. Overall, the findings suggest that school-based nutrition interventions, including nutrition education, food environment, those based on all three domains of the HPS framework, and eHealth interventions, can have a positive effect on some dietary outcomes, including fruit, fruit and vegetables combined, and fat intake. These results should be interpreted with caution, however, as the quality of the reviews was poor. Though these results support continued public health investment in school-based nutrition interventions to improve child dietary intake, the limitations of this umbrella review also highlight the need for a comprehensive and high quality systematic review of primary studies.

## 1. Introduction

Globally, poor diet is a leading preventable risk factor for the development of non-communicable diseases (NCDs), such as cardiovascular disease, specific types of cancer, and type 2 diabetes [1]. In 2019, an estimated 7.94 million deaths and 188 million disability-adjusted life-years were attributable to poor dietary intake [1]. As a result, promoting healthy diets at all life stages is a global public health priority [2].

Public health strategies to improve the diets in children are a recommended strategy to reduce the burden from NCDs, as dietary behaviours established in childhood often persist into adulthood [2,3,4]. Additionally a healthy diet is essential during childhood and adolescent years for optimal growth, health, and development [5,6,7]. For example, a healthy diet has been shown to enhance children’s cognitive skills such as concentration and memory, and improve mood, energy levels, and academic performance [8,9,10]. Conversely, consuming higher levels of unhealthy foods, such as fast food and sugar sweetened beverages (SSB), has been associated with behavioural problems, poor concentration, obesity, and emotional development problems [8,11,12].

Globally, population surveys indicate that children and adolescents do not meet dietary intake recommendations [5,13,14]. For example, data from the 2015 global school-based student health survey found 35% of children and adolescents aged 12 to 17 years consumed fruit less than once per day, and 21% consumed vegetables less than once per day [13]. In younger school-aged children, a recent survey (2017–2018) of Australian children reported that 72% and 4.4% of children aged 5 to 14 years, respectively, met the recommended guidelines for fruit and vegetable intake [14].

Schools are a key setting to influence children’s and adolescents’ healthy eating behaviours, given their existing infrastructure and estimates that children consume a third of their total energy intake during school hours [15]. Additionally, the school setting provides access to a large proportion of children for prolonged periods, and offers an opportunity to reduce population-wide chronic disease. Internationally, various governments and health organisations recommend schools implement policies and practices to create a school environment that supports students in making healthy choices [16,17,18]. For example, the World Health Organisation (WHO) recommends schools implement a whole school approach to healthy eating, and include strategies that target the school curriculum (i.e., learning, teaching, professional development), environment (i.e., physical, culture, policies, procedures), and partnerships (i.e., students, families, staff, community) [16].

Globally, there has been considerable investment in research to identify effective school-based nutrition interventions [19]. To identify which school-based nutrition interventions should be prioritised for investment, policymakers and practitioners require high quality synthesis of all available research evidence from the most robust trials. Several systematic reviews have been conducted of school-based nutrition interventions to provide this information [20,21,22,23,24,25,26]. However, such reviews synthesised studies of various study designs and intervention types [20,21,22,23,24,25,26]. For example, some reviews have synthesised randomised controlled trials (RCTs) with non-randomised trials [21,24,25], some reviews have synthesised school-based nutrition interventions with those targeting other health risk behaviours (e.g., smoking, alcohol use, sexual health) [20,22], and others have synthesised school-based interventions with those delivered in other settings, such as community and home [23,26]. Such syntheses, coupled with mixed findings from these reviews for different types of interventions, inhibits policy-makers’ ability to determine which approach is the most effective and should be prioritised for implementation. Umbrella reviews can help overcome this challenge efficiently and rapidly by consolidating the findings across several systematic reviews [27]. For example, a recently published umbrella review of school-based nutrition programs by the WHO synthesised any type of review (including scoping and literature reviews) that synthesised studies of any design (e.g., RCTs and non-randomised trials), implementing any intervention for diet-related outcomes up to 2019 [19]. This review did not, however, report the results of RCTs independently, and a number of systematic reviews have been published since [25,28,29,30]. An umbrella review of high-quality systematic reviews restricted to RCTs as the gold standard of effectiveness is required to update the evidence base regarding school-based nutrition interventions to inform current policy and practice.

The primary aim of this umbrella review was to summarise the evidence from systematic reviews that assess the effect of RCTs of school-based nutrition interventions designed to improve dietary intake outcomes in children aged 6 to 18 years. The secondary aims were to summarise any estimates of absolute costs or the cost-effectiveness of interventions and adverse effects reported in the included reviews.

## 2. Materials and Methods

The review was conducted following recommended methods outlined by the Cochrane Handbook for the conduct of overviews of systematic reviews [27]. The findings are reported in accordance with the Preferred Reporting Items for Systematic Reviews and Meta-analysis (PRISMA) checklist [31]. The review protocol was prospectively registered with PROSPERO (CRD42021218922), and the search strategy was deposited in a publicly available Open Science framework (https://osf.io/8t3sp, accessed on 17 November 2020) before conducting the search.

### 2.1. Study Inclusion and Exclusion

We included systematic reviews of RCTs (individual or cluster design) of school-based nutrition interventions which aimed to improve children’s dietary intake (either primary or secondary aim of included reviews), including systematic reviews of health economic evaluations. Based on Cochrane guidelines, we defined systematic reviews as reviews that report explicit reproducible methodology, i.e., included a comprehensive search with explicit criteria for including or excluding studies, and acceptable methods for assessing the quality of included studies (i.e., using an existing tool or reporting the criteria used to make assessments on the quality of the included studies) [27]. Reviews including a combination of RCTs and other study designs (e.g., non-randomised trials) were included if results were reported separately for RCTs. Reviews could include studies with any type of control group, e.g., usual practice, no intervention, attention control, or comparing alternative interventions (i.e., head-to-head comparisons). We excluded reviews of exclusive qualitative research, as they do not allow for estimation of effect sizes, which was the primary aim of the review. Reviews published prior to 2010 were excluded, as reviews published in the last 10 years were considered to capture the most updated synthesis of primary research.

Reviews were included if they reported studies of school-aged children and adolescents, aged between 6 and 18 years, that tested school-based nutrition interventions for improving children and/or adolescent dietary intake outcomes without restrictions on intervention mode of delivery, duration, or provider. Reviews were included if results were reported separately for children aged 6 to 18 years, or if the majority of the included studies targeted children and adolescents aged 6 to 18 years. Since school-based programs rarely focus on single health behaviours [19], reviews were included if they evaluated the effect of school-based interventions targeting multiple health behaviours (e.g., physical activity, sedentary behaviour, tobacco and alcohol use prevention) if nutrition was an explicit target of one or more of the intervention strategies, and if they reported results for dietary intake outcomes. Reviews encompassing multiple settings (e.g., school, home, community) were included if results were reported separately for the school-based studies.

The primary outcome for the umbrella review was objectively (e.g., weighed food observations) or subjectively (e.g., self-reported dietary assessment tools) measured child dietary intake. Typical examples of dietary intake outcomes include types and/or quantity of food consumed (e.g., fruits and vegetables, energy-dense nutrient-poor foods), change in diet quality (food indices), and types and/or quantity of beverages consumed (e.g., water, SSB). To be included, systematic reviews must have synthesised the effects of school-based interventions on the dietary intake outcome(s) of included studies either narratively or quantitatively (e.g., meta-analysis). We excluded nutrition-related measures that did not describe dietary intake (e.g., knowledge, attitudes, self-efficacy, or foods purchased or selected by children, or provided to children). Secondary outcomes included estimates of absolute costs or the cost-effectiveness of interventions, and any adverse effects reported in the included reviews. Adverse effects could include any physical, behavioural, psychological, or financial impact on the child, parent, or family, or the school where an intervention may have been implemented.

### 2.2. Search Strategy

The electronic databases Medline, Embase, CINAHL, the Cochrane database of systematic reviews, and Scopus were searched from 1 January 2010 to 20 June 2021 for relevant reviews published in the English language, using a combination of relevant keywords for participants, intervention, study design, and comparator consistent with the United States (US) National Library Medical Subject Headings (MeSH^®^) Thesaurus [32]. Based on recommendations from an information specialist (DB), the search strategy was expanded after protocol registration to include the Education Journals database, which was identified as an important database for education research. The search strategy was performed by DB, and modified to suit each database (see Appendix A for the detailed search strategy). We also searched all publications listed in the WHO eLibrary of Evidence for Nutrition Actions (eLENA) for any additional systematic reviews (no limits, up to 14 July 2021), and conducted targeted searches in: the systematic review repository Epistemonikos (From 2010 up to 22 June 2021); PROSPERO register (no limits, up to 22 June 2021); and GoogleScholar, examining the first 200 records (From 2010 up to 21 June 2021). To identify any additional potentially eligible reviews not picked up in the database or grey literature search, we screened all the reviews included in the reference list in two important overviews conducted by the WHO that included reviews of interest [19,33]. Finally, we searched the reference list of the reviews included in our umbrella review for additional reviews to screen. We imposed no restrictions by publication status.

### 2.3. Study Selection

We de-duplicated, uploaded, and screened records using Covidence software [34]. Pairs of review authors independently screened titles and abstracts and full text articles of all identified reviews in duplicate for eligibility (KO, CB, LD). Reasons for exclusion of reviews assessed at full text were recorded and are presented in Figure 1. Disagreements for title and abstract and full text screening were resolved by discussion and consensus, or by consultation with a third review author (RH).

As the aim of the umbrella review was to summarise the current body evidence of school-based nutrition interventions designed to improve dietary intake outcomes in children, all relevant systematic reviews, regardless of primary study overlap, were included [27]. However, if a review did not contribute any unique data to the umbrella review, i.e., all primary studies included in the review were included in other reviews, the review was excluded, as it did not contribute to the evidence synthesis [27]. An assessment of the degree of overlap in primary studies was undertaken as per Cochrane guidance [27] (see Appendix A).

### 2.4. Data Extraction

Data were extracted independently and in duplicate by two pairs of review authors (KO, EC, TD, SL, RW). To guide the extraction of data, a standardised tool was developed based on recommendations from the Joanna Briggs Institute [35]. The tool was piloted by the review team and subsequently revised before use. Consistent with the Cochrane methodology for overviews of systematic reviews, data extraction was limited to what was presented by the included systematic review, i.e., no data from the primary studies included in the reviews were extracted [27]. Information extracted from each review included review characteristics: citation details; objectives; search details; selection criteria (e.g., population, intervention, comparator, outcome, study designs); methods of appraising the quality of primary studies included; data synthesis and analysis; and results, including the number of primary studies included, description of included interventions and comparators, results of quality assessments, dietary intake outcomes and synthesised effects of interventions, and funding support for the reviews. Disagreements were resolved by discussion and consensus, or by consultation with a third review author (RH).

### 2.5. Assessment of Methodological Quality

Two independent review authors (AB, FS) assessed the overall quality of included reviews using the Assessing the Methodology Quality of Systematic Reviews tool 2 (AMSTAR2) [36]. The tool includes 16 domains relating to the research question, review design, search strategy, study selection, data extraction, and justification for excluded studies, description of included studies, risk of bias, sources of funding, meta-analysis, heterogeneity, publication bias, and conflicts of interest. Review authors were required to select ‘yes’, ‘partial yes’, ‘no’, or ‘no meta-analysis conducted’ (as applicable) for each of the 16 domains. Disagreements were resolved by discussion and consensus, or by consultation with a third reviewer (KO). As per AMSTAR2 guidance, critical domains should be identified for each umbrella review (i.e., based on domains that are most important for the included reviews under consideration) and weighted more heavily when rating the overall confidence in the results of the review, due to their greater effect on the validity of the review findings [36]. Based on guidance, we identified six critical domains: protocol registration; adequacy of literature search; performing risk of bias assessment; appropriateness of meta-analytical method; consideration of risk of bias when interpreting the results; and assessment of publication bias [36]. AMSTAR2 quality assessments of the included reviews are presented in tabular form for each review (see Appendix A).

### 2.6. Data Synthesis

Where available, we report the effect estimates, measures of variance, and heterogeneity of studies quantified in meta-analysis. Where meta-analysis was not available, we summarised the findings of the narrative synthesis. Where reviews included both an assessment of risk of bias or quality of primary included studies (e.g., Cochrane risk of bias tool) [37], and an evidence grading system, such as Grading of Recommendations, Assessment, Development and Evaluation (GRADE) [38], we reported the findings from the grading system, as GRADE considers both quality assessments of the primary studies, as well as the quality of the evidence included in the review overall [39].

We undertook a narrative synthesis of reviews, organising the results of the reviews by intervention type and dietary intake outcome. The types of interventions synthesised in each review were categorised according to one or all of the three WHO Health Promoting Schools (HPS) framework domains [22]: nutrition education; food environment; all three HPS framework domains (i.e., all primary studies had to include education, environment, and partnerships); and other. School-based nutrition interventions categorised as ‘other’ were those where the original review author did not restrict the inclusion, or report the results, by a particular type of intervention or HPS framework domain. Two review authors (KO, RH) determined the type of interventions based on intervention descriptions provided in the reviews, and disagreements were resolved by discussion and consensus.

We categorised the findings of each review for each dietary intake outcome according to effectiveness based on a framework previously applied in a Cochrane overview of reviews [40]. The framework categorises interventions as follows:Likely effective: indicating that the review found evidence of effectiveness for an intervention (if meta-analysis found an effect, or if all included primary studies were effective in narrative synthesis, as described by the authors of included reviews).Promising (more evidence needed): indicating that the review found some evidence of effectiveness for an intervention, but that more evidence is needed (if the majority (>50%) of the included primary studies in narrative synthesis were effective, as described by the authors of included reviews).Ineffective: indicating that the review found evidence of lack of effectiveness for an intervention (if meta-analysis did not find an effect, or if all included primary studies in narrative synthesis were ineffective, as described by the authors of included reviews).Probably ineffective (more evidence needed): indicating that the review found evidence suggesting lack of effectiveness for an intervention, but more evidence is needed (if the majority (>50%) of included primary studies in narrative synthesis were ineffective, as described by the authors of included reviews).No conclusions possible due to lack of evidence: indicating that the review found insufficient evidence for review authors to comment on the effectiveness of an intervention (where only one primary study included in the review measured a particular dietary intake outcome, as described by the authors of included reviews).

If a review included a meta-analysis and narrative synthesis for the same dietary intake outcome, a rating for both analyses was included. Three review authors (KO, RH, SY) independently categorised the reviews using the above framework based on the reported results and conclusions of the original reviews. Disagreements were resolved by discussion and consensus.

## 3. Results

A total of 8570 records were identified through database searching, and 1799 through other sources. After de-duplication, 9257 were screened at title and abstract, and 8978 were excluded. Of the 279 full text articles screened, 258 were excluded, and 16 reviews (21 articles) were included. After assessing the potential overlap of primary studies in the included reviews, three reviews were excluded due to not contributing any unique primary studies to the umbrella synthesis [41,42,43] (see Appendix A for the overlap of the primary studies of the included reviews). Thirteen reviews were included in the review (see Figure 1).

### 3.1. Characteristics of Included Reviews

Characteristics of the 13 included reviews overall (i.e., all primary studies reported in the review irrespective of study design and other study characteristics) are described in Table 1. Of these, two reviews synthesised results for nutrition education interventions [26,44], three for food environment interventions interventions [24,25,45], one for HPS nutrition interventions [22], and seven synthesised other types of nutrition interventions [20,21,23,28,29,30,46]. Included reviews were published between 2011 and 2021. The number of primary studies included within reviews ranged from 10 to 100. Of the 13 reviews, three reviews restricted the inclusion criteria for country, and one review each included only primary studies from developed countries [44], the United Kingdom (UK) and Europe [29], and middle-income countries [30]. Eleven reviews focused on interventions conducted solely in school-based settings [20,21,22,24,25,28,29,30,44,45,46], and two reviews included interventions conducted across multiple settings (e.g., school, home, community), where results for the school setting were reported separately [23,26]. Of the 13 reviews, seven reviews only included RCTs (individual or cluster) as part of their eligibility criteria [20,22,23,26,28,30,44], whereas six included a range of study designs (e.g., RCTs and non-randomised trials), where results for the RCTs were reported separately [21,24,25,45,46,47]. Three reviews included children of any age up to 18 years [23,24,25], three reviews included children aged between 4 and 18 years [22,26,45], four included children aged between 4 and 12 years [21,28,30,46], and three included children aged between 10 and 18 years [20,44,47].

Of the 13 included reviews, three did not report whether funding support was received to undertake the review [30,46,47], two reviews reported no funding support [21,44], and the remaining eight reviews reported funding support from government or charitable organisations [20,22,23,24,25,26,28,45].

### 3.2. Characteristics of Included Reviews Relating to RCTs of School-Based Nutrition Interventions

The characteristics of the RCTs of school-based nutrition interventions for dietary intake outcomes within included reviews are shown in Table 1. All reviews compared the effectiveness of school-based nutrition interventions to a control group (i.e., usual practice, no intervention, attention control), and one review also compared alternative interventions [46]. The number of RCTs synthesised across the reviews ranged from two to 20. The duration of the interventions ranged from a single day to seven years. Intervention delivery personnel were reported in three reviews [30,44,46], and included teachers/school staff, researchers, and health professionals (e.g., dietitians).

Across the 11 reviews reporting which studies were included in the synthesis, there were 82 unique primary RCTs of school-based nutrition interventions for dietary intake outcomes (see Appendix A). Of the 82 primary studies, 61 (74%) primary studies were included in a single review, and 21 (26%) primary studies were included across two or more reviews. Fifty-six primary studies were cluster RCTs (CRCTs), nine were RCTs, and for the remaining 17 RCTs, it was unclear if they were individual or cluster designs. Two reviews did not provide enough information to be able to identify which of the primary studies were included in the meta-analyses [21,28]. The majority of primary studies were undertaken in North America (44% (36/82), 31 in the US, three in Mexico, two in Canada) and Europe (41% (34/82), with the majority in the UK, the Netherlands, and Norway (*n* = 7 each)), with four in Asia, three each in South America and the Middle East, and two in Australasia. The number of participants included in primary studies ranged from 91 to 4603 students.

Most reviews examined the effectiveness of nutrition interventions on more than one dietary intake outcome (77%, 10/13) [20,22,23,24,25,28,29,30,44,45]. Nine reviews reported fruit and/or vegetable intake [20,21,22,24,25,28,29,44,46], five reviews reported fat intake [20,22,24,44,47], four reviews reported intake of SSB [24,26,44,47], three reviews reported dietary intake in general (i.e., no specific foods, nutrients) [23,44,45], two reviews each reported unhealthy snack intake [20,24] and total energy intake [28,29], and one review each reported intake of fish [44], fibre [44], water [44], and sucrose [44], as well as daily caloric intake (kcal) [24] and frequency of breakfast intake [29].

Seven reviews synthesised the results using meta-analyses only [20,21,22,23,24,25,28], two included both meta-analyses and narrative synthesis [26,46], and four reviews only described the results narratively [30,44,45,47].

### 3.3. Methodological Quality Assessment of Included Reviews

Of the 13 reviews, two were rated as high quality (15%) [22,23], one as moderate quality (8%) [30], three as low quality (23%) [20,45,47], and seven as critically low quality (54%) [21,24,25,26,28,44,46]. The ten reviews that were assessed as low or critically low were downgraded due to various critical flaws, the most common of which was failing to discuss the possible impact of risk of bias when interpreting/discussing the results of the review (six reviews). Other common reasons for downgrading the evidence included not explicitly stating review methods were established a priori (four reviews), not using a comprehensive literature search strategy (four reviews), failing to use a satisfactory technique for assessing the risk of bias in individual studies (four reviews), and not carrying out an adequate investigation of publication bias (small study bias) and discussing its likely impact on the results of the review (three reviews). The one review that was rated moderate quality did not have any critical flaws, but was downgraded due to various critical weaknesses, for example, not providing a satisfactory explanation for, and discussion of, any heterogeneity observed in the results of the review. Further details of the critical appraisal can be found in Appendix A.

### 3.4. Effect of Interventions

Table 2 reports summary results of the included reviews according to the effectiveness categorisation framework, and Table 3 summarises the main findings of the umbrella review. Both tables are organised by type of intervention. All results are reported for intervention compared to control.

#### 3.4.1. School-Based Nutrition Education Interventions (*n* = 2 Reviews, 1 Review Meta-Analysis and Narrative Synthesis, 1 Reviews Narrative Synthesis)

We identified two reviews that assessed the effectiveness of school-based nutrition education interventions on improving dietary intake outcomes [26,44]. Across the two reviews, results were reported by educational and behavioural interventions [26], and multistrategy interventions encompassing nutrition education and other complementary strategies [44]. One (critically low quality) review of 12 RCTs of educational and behavioural nutrition interventions reported no overall effect on SSB intake in meta-analysis of two trials (mean difference (MD) −26.53 95% confidence interval (CI): −53.72 to 0.66, I^2^ = 6%, 2914 participants), and mixed effects in narrative synthesis for the remaining ten trials [26]. One (critically low quality) review of 12 RCTs of multistrategy interventions in developed countries (e.g., the US, Europe, Australia) encompassing nutrition education and other complementary strategies to improve diet (e.g., changes in school nutrition policies) reported a positive effect in narrative synthesis in reducing fat intake (five trials) and improving fish intake (one trial), and mixed findings for fruit (five trials), fruit and vegetables combined (five trials), and SSB intake (three trials) [44] (see Table 3).

After applying the effectiveness categorisation framework, we found school-based nutrition education interventions are likely to be effective for reducing fat intake [44] (one review), and promising for increasing fruit intake [44] (one review). Mixed findings were found for SSB intake, with one review categorised as probably ineffective [44], and the second review categorised as promising (based on narrative findings, 10 trials) and ineffective (based on meta-analysis, two trials) [26]. School-based nutrition education interventions are probably ineffective for fruit and vegetable intake combined [44] (one review). No conclusions could be drawn for fish intake due to the lack of evidence to date (i.e., data only reported for one individual study within one review) [44] (see Table 2).

#### 3.4.2. School Food Environment Interventions (*n* = 3 Reviews, 2 Reviews Meta-Analysis, 1 Review Narrative Synthesis)

We found three reviews that assessed the effectiveness of school food environment interventions on improving dietary intake outcomes [24,25,45]. Across the three reviews, results were reported by school food environment policy interventions [24], school environment interventions [25], and school environment interventions without education or school-based health services [45]. One (critically low quality) review of 12 RCTs of school food environment policy interventions targeting food and beverage availability, including the provision of healthful foods and improving the nutritional quality standards for school meals and competitive foods and beverages, reported a positive effect in meta-analysis for fruit (effect size (ES) 0.27 95%CI: 0.09 to 0.45, I^2^ = 50%, six trials), fruit and vegetables combined (ES 0.37 95%CI: 0.05 to 0.69, I^2^ = 35%, six trials), fat (ES −7.15, 95%CI: −11.36 to −2.95, I^2^ = 89%, two trials), saturated fat (ES −2.74, 95%CI: −4.99 to −0.48, I^2^ = 90%, two trials), SSB (ES −0.02 95%CI: −0.03 to −0.01, 1 trial), unhealthy snacks (ES −0.06, 95%CI: −0.09 to −0.02, I^2^ = 0%, two trials), and daily caloric intake (kcal) (ES −58 95%CI: −84 to −33, two trials), but no effect for vegetable intake (ES 0.02 95%CI −0.25 to 0.29, I^2^ = 7%, three trials) [24]. Similarly, one (critically low quality) review of 14 RCTs of school environment interventions, including programs, strategies, and policies that aimed to modify infrastructure and conditions, found interventions were effective for fruit intake (MD 0.2 95%CI: 0.14 to 0.26, I^2^ = 67%, 10 trials), but were not effective for vegetable intake (MD 0.00 95%CI: −0.01 to 0.01, I^2^ = 69%, seven trials) [25]. One (low quality) review of two RCTs of school environment interventions without education or school-based health services reported no effect in narrative synthesis for improving diet in general [45] (see Table 3).

After applying the effectiveness categorisation framework, we found school food environment interventions are likely to be effective for improving fruit [24,25] (two reviews), fruit and vegetables combined [24] (one review), fat [24] (one review), saturated fat [24] (one review), unhealthy snacks [24] (one review), and caloric intake (kcal) [24] (one review), but are ineffective for increasing vegetable intake [24,25] (two reviews) and diet in general [45] (one review). No conclusions could be drawn for intake of SSB due to the lack of evidence to date (i.e., data only reported for one individual study within one review) [24] (see Table 2).

#### 3.4.3. School-Based Nutrition Interventions Based on the HPS Framework (*n* = 1 Review, Meta-Analysis)

One (high quality) review reported the effectiveness of interventions that adopted all domains of the HPS framework in improving dietary intake [22]. Pooled analysis found nutrition-only HPS interventions were effective for increasing fruit and vegetable intake combined (standardised mean difference (SMD) 0.15 95%: CI 0.02 to 0.29, I^2^ = 83%, nine trials, 6210 participants), but were not effective for reducing fat intake (SMD −0.08 95%CI: −0.21 to 0.05, I^2^ = 68%, seven trials, 4216 participants). Nutrition and physical activity HPS interventions were not found to be effective for improving fruit and vegetables combined (SMD 0.04 95%CI: −0.18 to 0.26, I^2^ = 79%, four trials, 6612 participants), or fat intake (SMD −0.04 95%CI: −0.20 to 0.12, I^2^ = 95%, 10 trials, 12,460 participants) [22] (see Table 3).

After applying the effectiveness categorisation framework, we found nutrition-only HPS interventions are likely to be effective for improving fruit and vegetable intake combined, but are ineffective for fat intake, and nutrition and physical activity HPS interventions are ineffective for fruit and vegetables combined, and fat intake (see Table 2).

#### 3.4.4. Other Types of School-Based Nutrition Interventions (*n* = 7 Reviews, 5 Reviews Meta-Analysis, and 2 Reviews Narrative Synthesis)

We found seven reviews that assessed the impact of other types of school-based nutrition interventions for improving dietary intake outcomes that couldn’t be categorized against the HPS framework [20,21,23,28,29,30,46]. Across the seven reviews, results were reported by any school-based nutrition intervention [21], school-based nutrition interventions targeting multiple health behaviours (e.g., diet, physical activity, tobacco use) [20,23], one delivered primarily via electronic Health (eHealth) methods (e.g., internet, computers) [20], any school-based nutrition intervention synthesised by eHealth interventions, free or subsidised fruit and vegetable distribution interventions, and multistrategy interventions (e.g., curriculum, environment, parent and teacher involvement) [46], school-based obesity prevention interventions (i.e., interventions targeting both nutrition and physical activity) [28,30], and European school food interventions [29].

One (critically low quality) review of nine RCTs of any school-based nutrition interventions reported no effect on fruit and vegetable intake combined in meta-analysis of four trials (ES 0.26 95%CI: 0.12 to 0.40, I^2^ = 1%) [21]. One (high quality) review of three RCTs of school-based interventions targeting nutrition, physical activity, and tobacco use found no overall effect on improving diet in general in meta-analysis of three trials (odds ratio (OR) 0.82 95%CI: 0.64 to 1.06, I^2^ = 49%, 6441 participants) [23]. One (low quality) review of seven RCTs of eHealth school-based interventions targeting multiple health behaviours reported eHealth interventions were effective for fruit and vegetables intake combined immediately after the intervention (SMD 0.11 95%CI: 0.03 to 0.19, I^2^ = 42%, six trials, 7390 participants), but not effective at follow-up (SMD 0.07 95%CI: −0.01 to 0.15, I^2^ = 52%, three trials, 6004 participants, follow-up range: 4 months to 36 months) or for fat (SMD −0.06 95%CI: −0.15 to 0.03, I^2^ = 52%, three trials, 5240 participants), or intake of unhealthy snacks (including SSB) immediately after the intervention (SMD −0.02 95%CI: −0.10 to 0.06, I^2^ = 54%, three trials, 5812 participants) or at follow-up (SMD −0.06 95%CI: −0.15 to 0.0, I^2^ = 17%, two trials, 2667 participants, follow-up range: 4 months to 24 months) [20]. One (critically low quality) review of 11 RCTs of any school-based nutrition intervention for improving fruit and vegetables combined intake found eHealth (e.g., computers) interventions were effective (SMD 0.33 95%CI: 0.16 to 0.50, I^2^ = 0%, two trials, 606 participants), but free or subsidised fruit and vegetable distribution interventions (SMD 0.02 95%CI: −0.08 to 0.12, I^2^ = 0%, two trials, 1536 participants), and multistrategy interventions (SMD 0.08 95%CI: −0.00 to 0.17, I^2^ = 50%, seven trials, 4800 participants) were not [46]. One (critically low quality review) of 11 RCTs of school-based obesity preventions interventions reported interventions were not effective for total energy intake (MD 5.23 95%CI: −77.83 to 88.28, four trials, 1576 participants), or fruit and vegetable intake measured in portions per day (MD 0.05 95%CI: −0.08 to 0.17, five trials, 4741 participants) and grams per day (MD 10.45 95%CI: −17.53 to 38.43, two trials) [28]. One (moderate quality) review of 10 RCTs of school-based obesity preventions interventions in middle-income countries reported mixed effects in narrative synthesis for improving diet in general [30]. One (low quality) review of nine RCTs of European school food interventions reported positive effects in narrative synthesis in reducing SSB intake (three trials), saturated fat (one trial), total energy intake (one trial), and improving breakfast intake (one trial), and mixed findings for improving fruit and vegetable intake combined (six trials) [29] (see Table 3).

After applying the effectiveness categorisation framework, we found evidence that school-based nutrition interventions, including those delivered via eHealth, are likely to be effective for increasing fruit and vegetable intake combined [20,21,46] (three reviews), and European food interventions broadly to be effective for SSB [29] (one review) and promising for fruit and vegetable intake combined [29] (one review). Promising evidence was also found for the effectiveness of obesity prevention interventions in improving diet in general [30] (one review). School-based nutrition interventions found to be ineffective included: eHealth interventions for fat [20] (one review) and unhealthy snack intake [20] (one review); free or subsidised fruit and vegetable distribution interventions for fruit and vegetable intake combined [46] (one review); multistrategy interventions for fruit and vegetable intake combined [46] (one review); school-based health intervention targeting multiple behaviours for improving diet in general [23] (one review); and school-based obesity prevention interventions for intake of fruit and vegetables combined, and total energy [28] (one review). No conclusions could be drawn for intake of saturated fat and breakfast frequency due to the lack of evidence to date (i.e., data only reported for one individual study within one review) [29] (see Table 2).

### 3.5. Intervention Costs

Five reviews intended to synthesise intervention costs if the included primary studies reported any cost data [22,23,24,25,30]. Four reviews reported that costs data were generally not reported, and no synthesis of cost data was included in the reviews [23,24,25,30]. One review reported that only one of the 20 RCTs of school-based nutrition interventions provided cost data which reported that the estimated costs of the intervention were “GBP 378 for capital and development costs plus GBP 13.50 consumables per school” [22].

### 3.6. Adverse Events

Four reviews intended to synthesise adverse events or unintended adverse consequences of included primary studies [22,23,25,30]. One review reported that three of the 23 RCTs of school-based nutrition intervention measured adverse events, all of which reported no adverse events [22]. One review reported that four of the 10 RCTs of school-based nutrition intervention measured adverse events, however these were not described by the review authors [30]. The remaining two reviews did not report any data regarding the number or findings of included studies that assessed adverse events [23,25].

## 4. Discussion

### 4.1. Summary of Main Results

The purpose of this umbrella review was to summarise the evidence for school-based nutrition interventions designed to improve dietary intake outcomes in children aged 6 to 18 years. Though there was considerable heterogeneity in the intervention approaches synthesised, the outcomes and findings reported across reviews, overall, suggest that school-based nutrition interventions can have a positive effect on some dietary intake outcomes. Broadly, intervention approaches reported to be effective on at least one dietary intake outcome included nutrition education, food environment, those based on all three domains of the HPS framework, and eHealth interventions, however positive results were only found for intake of fruit, fruit and vegetables combined, or fat, and the quality of reviews was generally poor.

Across reviews, there was consistent or uncontested evidence of the likely effectiveness that: school-based nutrition education interventions reduce fat intake; food environment interventions improve fruit, fruit and vegetables combined, fat, unhealthy snacks, and caloric intake; and nutrition interventions adopting the HPS framework improve fruit and vegetable intake combined. Of the other school-based nutrition interventions synthesised in reviews, eHealth interventions are likely to improve fruit and vegetable intake combined, and European school-based nutrition interventions are likely to improve SSB intake. However, the majority of these positive results were reported by low quality reviews, as assessed by AMSTAR2, and the only positive results of the two reviews assessed as high quality were for nutrition interventions adopting the HPS framework, effective for improving fruit and vegetable intake combined.

### 4.2. Overall Completeness and Applicability of Evidence

The umbrella review did not restrict the types of school-based nutrition interventions to be included, providing a valuable representation of all the systematic reviews of the effectiveness of school-based nutrition interventions for improving dietary intake. However, due to the high level of variability in the study populations, interventions and outcomes of included reviews, and how these were synthesised, these findings should be interpreted with caution.

Some of the methods and selection criteria for our review may, however, have biased the results. For example, we only included reviews that synthesised the results of RCTs separately. As a result, a number of reviews that synthesised only non-randomised controlled and uncontrolled study designs, or synthesised these with RCTs, were excluded. Second, some reviews were not able to synthesise all their included studies in their findings. This may have resulted in some eligible primary studies not being synthesised in this umbrella review, as data extraction was limited to the reported synthesised findings from each eligible review.

The results of the review are likely generalisable to middle- and high-income countries, on the basis that the primary studies included within reviews were all implemented in middle- and high-income countries. The generalisability and external validity to low-income countries is, however, limited, and more primary study research is needed in low-income countries.

The evidence regarding the cost-effectiveness and adverse consequences of the interventions synthesised in existing systematic reviews is also limited. Only one of the included reviews attempted to synthesise trial information about intervention costs, and two reviews on any unintended adverse consequences of the intervention. Though it is unclear whether the omission to synthesise this information at the review level was due to the likely absence of such data in individual studies, such information is required by practitioners and policymakers to prioritise effective interventions. These factors are important considerations for practitioners and policymakers, but are often not reported in RCTs [52], or examined in systematic reviews [53].

### 4.3. Quality of the Evidence

The various risk of bias and quality assessment tools used in included reviews precluded the ability to synthesise this information across primary studies. However, the majority of individual studies were assessed as moderate to low quality.

Similarly, the methodological quality of the majority of included reviews was assessed as low to critically low, with only two of the 13 reviews assessed as high quality, and one as moderate quality. Such assessments highlight the need for the conduct of high quality primary studies, and systematic reviews of school-based nutrition interventions. This is particularly concerning, given the need for policymakers and practitioners to have access to reliable and summarised data to inform school-based nutrition policies and practices. Better adherence to the established guidelines for conducting and reporting for systematic reviews is needed to ensure the rigour, validity, and reliability of the review results.

### 4.4. Agreements and Disagreements with Other Studies or Reviews

The results of this umbrella review are generally consistent with those reported in a recent WHO umbrella review [19]. For example, the WHO review reported positive effects on diet-related outcomes for nutrition education interventions, creating healthy environments for nutrition, and implementing comprehensive school nutrition policies targeting multiple components (e.g., diet and physical activity) and multiple approaches (e.g., education and environment) [19]. However, the WHO review did not report the results of interventions on specific dietary intake outcomes (e.g., fruit intake), so we are unable to make any comparisons by outcome [19].

### 4.5. Strengths and Limitations of the Umbrella Review

The review provides the most rigorous, comprehensive, and current synthesis of evidence of school-based nutrition interventions for improving student diet. The review adopted Cochrane methodology and guidance for the conduct of overviews of systematic reviews, including duplicate independent assessment of eligibility and data extraction, the methods were registered prospectively with PROSPERO, and the search strategy was deposited with the Open Science framework before conducting the search. We also undertook a comprehensive search without publication restrictions.

The findings of the review should be considered in the context of its limitations. First, there was a high level of variability amongst studies in the included reviews, which limits the ability to determine which intervention approaches are effective when implemented as part of a multistrategy intervention. Second, systematic reviews of obesity prevention interventions where the primary outcome was weight-related, e.g., body mass index, were included if a dietary intake outcome was synthesised by the review author. This means that some reviews excluded papers that did not report a weight outcome, and, therefore, may have excluded data for diet outcomes relevant to our umbrella review. Third, although we excluded reviews that did not provide any unique data based on Cochrane best practice methods, many of the included reviews relied on the same primary studies. However, the overlap between reviews based on outcome, overall, appears to be minimal. Of the 82 primary studies included, only 17 studies (21%) were included in more than one review when reporting the same dietary intake outcome, and the majority of these (76%) were included in two reviews. Fourth, as a deviation from the registered methods, we made the post hoc decision to limit to systematic reviews published in the English language only. Lastly, we amended the eligibility criteria to only include dietary intake outcomes, as we deemed this the best outcome prior to undertaking full text screening. As a result, reviews that only reported nutrition-related knowledge were excluded. This post hoc decision was not deemed as a potential bias, given the focus of the review was establishing the types of interventions that were effective in improving child dietary intake outcomes.

## 5. Conclusions

School-based nutrition interventions can have a positive effect on dietary intake in children aged 6 to 18 years. In particular, school-based interventions, including nutrition education, food environment, those based on all three domains of the HPS framework, and eHealth interventions, appear to offer the most promise for improving intake of fruit, fruit and vegetables combined, and fat. Though these results support continued public health investment in school-based nutrition intervention to improve child dietary intake, the limitations of this umbrella review also highlight the need for a comprehensive and high quality systematic review of primary studies. Such a review would allow an opportunity to consolidate the evidence from all types of school-based nutrition interventions, and investigate the effectiveness of interventions overall, as well as their components.

## Figures and Tables

**Figure 1 nutrients-13-04113-f001:**
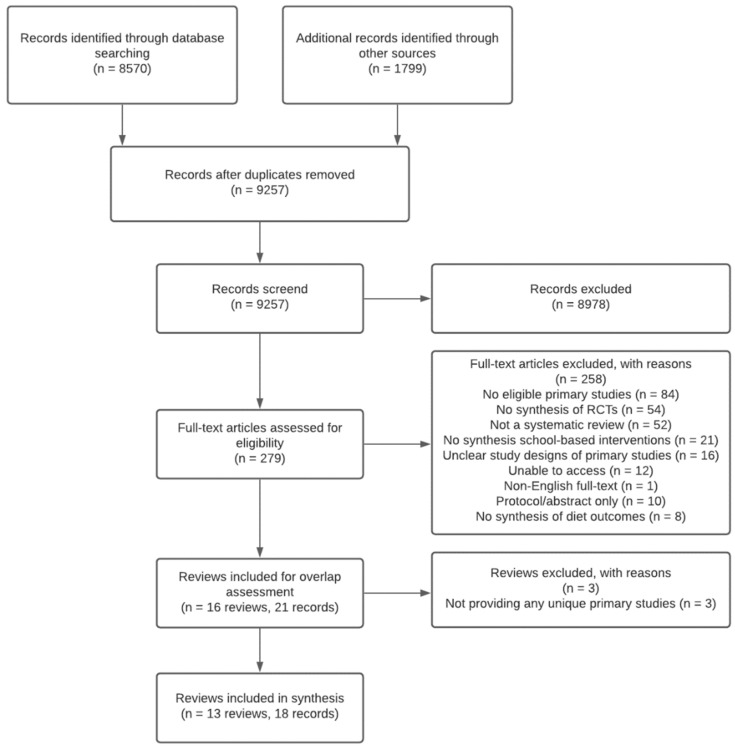
PRISMA diagram of the flow of included studies.

**Table 1 nutrients-13-04113-t001:** Characteristics of included systematic reviews (*n* = 13).

Author Year Country	Population (Years) Setting Design	Intervention (Duration) Comparator	Number of RCTs of School-Based Nutrition Interventions Reporting Dietary Intake Outcomes; Year of Publication; Countries Total Number of Primary Studies Included in Each Review	Dietary Intake Outcome (s)
School-based nutrition education interventions				
Meiklejohn 2016 [44] Country: Developed	Age: 10 to 18 Setting: School Design: RCTs	Intervention: Multi-strategy school-based nutrition education interventions on health and nutrition (NR, duration varied) Comparator: NR	4 RCTs (unclear if individual or cluster design), 8 CRCTs; 2003 to 2009; US (3), Belgium (2), Norway (2), Australia (1), Finland (1), Greece (1), Sweden (1), Netherlands, Norway and Spain (1) Total included: 13 studies	Fruit, FV, fat, SSB, fish, diet, sucrose, water, fibre
Rahman 2017 [26] Country: Any	Age: 4 to 16 Setting: School, home, community Design: RCTs	Intervention: Educational and behavioural interventions to reduce SSB intake (range: 10 weeks to 18 months) Comparator: No intervention	5 RCTs (unclear if individual or cluster design), 7 CRCTs; 2004 to 2014; Germany (3), Netherlands (2), Brazil (2), US (1), UK (1), Norway (1), Belgium (1), Portugal (1) Total included: 16 studies	SSB
School food environment interventions				
Bonell 2013 [45] Country: Any	Age: 4 to 18 Setting: School Design: prospective experimental, quasi-experimental	Intervention: School environment interventions that do not include health education or school-based health services for improving health and wellbeing (NR) Comparator: standard school practices	2 RCTs; 2003 to 2009; US (2) Total included: 10 studies	Diet in general
Micha 2018 [24] Country: Any	Age: 2 to 18 Setting: School Design: RCTs, quasi-experimental	Intervention: School food environment policies on dietary habits, adiposity, and metabolic risk (range: 2.3 to 33 months *) Comparator: NR	4 RCTs (unclear if individual or cluster design), 8 CRCTs; 1996 to 2012; US (4), Norway (2), UK (2), Canada (1), Finland (1), New Zealand (1), Netherlands, Norway and Spain (1) Total included: 91 studies	Fruit, vegetable, FV, fat, saturated fat, SSB, unhealthy snacks, daily caloric intake (kcal)
Pineda 2021 [25,48] Country: Any	Age: ≤19 Setting: School Design: and RCTs, CRCTs, quasi-experimental	Intervention: School food environment interventions to improve diet and prevent obesity (range: 5 weeks to 7 years) Comparator: No intervention, or a comparison of the same group before the implementation of the intervention	9 RCTs, 5 CRCTs; 1999 to 2015; US (5), UK (3), Norway (2), Canada (1), Denmark (1), New Zealand (1), Netherlands, Norway and Spain (1) Total included: 100 studies	Fruit, vegetable
School-based nutrition interventions based on the HPS framework				
Langford 2014 [22,49] Country: Any	Age: 4 to 18 Setting: School, college Design: CRCTs	Intervention: School interventions based on the HPS framework, with active components in school education, environment, and partnerships for improving health and wellbeing (range: 5 months to 3 years) Comparator: No intervention, usual practice, or an alternative intervention that included only one or two of the HPS criteria	20 CRCTs; 1998 to 2013; US (11), UK (3), Mexico (2), Belgium (1), Finland (1), Norway (1), Netherlands, Norway and Spain (1) Total included: 67 studies	FV, fat
Other school-based nutrition interventions				
Champion 2019 [20,50] Country: Any	Age: 11 to 18 Setting: School Design: RCTs, CRCTs	Intervention: School-based eHealth interventions targeting multiple health behaviours (range: 1 day to 36 months) Comparator: no intervention, education as usual, or an alternate evidence-based intervention not delivered via eHealth	1 RCT, 6 CRCTs; 2004 to 2013; US (4), Belgium (2), Netherlands (1) Total included: 16 studies	FV, fat, unhealthy snacks (including SSB)
Delgado-Noguera 2011 [46] Country: Any	Age: 5 to 12 Setting: School Design: RCTs, CRCTs, CCTs	Intervention: School-based nutrition interventions for promoting the intake of FV (range: 5 weeks to 3 years) Comparator: NR in criteria	13 CRCTs; 1998 to 2008; US (5), UK (2), Italy (2), Netherlands (2), Norway (1), Netherlands, Norway and Spain (1) Total included: 19 studies	FV
Evans 2012 [21,51] Country: Any	Age: 5 to 12 Setting: School Design: Randomised and non-randomised controlled trials	Intervention: School-based nutrition interventions on FV intake (range: 3 months to 2 years *) Comparator: Control or usual practice	9 RCTs (unclear if individual or cluster design) * Total included: 27 studies	FV
MacArthur 2018 [23] Country: Any	Age: Up to 18 Setting: School, home, community, clinic Design: RCTs, CRCTs	Intervention: Health interventions targeting multiple health behaviours (range: 9 months to 5 years) Comparator: Receiving usual practice, no intervention, or placebo or attention control	3 CRCTs; 1989 to 2015; US (2), India (1) Total included: 70 studies	Diet in general
Nally 2021 [28] Country: Any	Age: 5 to 12 Setting: School Design: RCTs, CRCTs	Intervention: School-based obesity prevention interventions (range: 12 weeks to 4 years *) Comparator: No intervention, alternative treatment condition or usual practice, i.e., existing physical education programme	11 RCTs (unclear if individual or cluster design) * Total included: 48 studies	FV, total energy intake
Rose 2021 [29,47] Country: UK and Europe	Age: 11 to 18 Setting: School Design: No restrictions	Intervention: European school food interventions on nutrition, weight, and wellbeing (NR) Comparator: NR	5 RCTs (unclear if individual or cluster design), 4 CRCTs; 2009 to 2017; Netherlands (3), Italy (2), Finland (1), Greece (1), Spain (1), UK (1) Total included: 27 studies	FV, saturated fat, SSB, total energy intake, breakfast frequency
Singhal 2020 [30] Country: Middle- income countries	Age: 4 to 12 years Setting: School Design: CRCTs	Intervention: School-based obesity prevention intervention (range: 3 to 36 months) Comparator: No intervention, usual practice, or an intervention with no specific diet or PA content	10 CRCTs; 2009 to 2018; Brazil (3), China (3), Mexico (1), Iran (1), Lenanon (1), Turkey (1) Total included: 21 studies	Diet in general

Abbreviations: CCT, clinical controlled trial; CRCT, cluster randomised controlled trial; eHealth, electronic health; FV, fruit and vegetables combined; HPS, Health Promoting Schools; MA, meta-analysis; NR, not reported; PA, physical activity; RCTs, randomised controlled trial; SSB, sugar-sweetened beverages; UK, United Kingdom; US, United States. * Evans 2012 and Nally 2021 did not provide enough information to be able to identify which of the primary studies were included in the meta-analyses.

**Table 2 nutrients-13-04113-t002:** Results of RCTs of school-based nutrition interventions compared to control according to the effectiveness categorisation framework *.

Author Year	Synthesis	Fruit	Vegetable	FV	Fat	Saturated Fat	SSB	Unhealthy Snacks	Calories	Diet in General	Total Energy	Fish	Breakfast
School-based nutrition education interventions													
Meiklejohn 2016 [44]	Narrative	Promising −3/5 trials		Probably ineffective −2/5 trials	Likely effective −5/5 trials		Probably ineffective −1/3 trials					No conclusions −1 trial	
Rahman 2017 [26]	MA and narrative						MA Ineffective −2 trials						
							Narrative Promising −6/10 trials						
School food environment interventions													
Bonell 2013 [45]	Narrative									Ineffective −0/2 trials			
Micha 2018 [24]	MA	Likely effective −6 trials	Ineffective −3 trials	Likely effective −6 trials	Likely effective −2 trials	Likely effective −2 trials	No conclusions −1 trial	Likely effective −2 trials	Likely effective −2 trials				
Pineda 2021 [25,48]	MA	Likely effective −10 trials	Ineffective −7 trials										
School-based nutrition interventions based on the HPS framework **													
Langford 2014 [22,49]	MA			Nutrition only Likely effective −9 trials	Nutrition only Ineffective −7 trials								
				NPA Ineffective −4 trials	NPA Ineffective −10 trials								
Other school-based nutrition interventions													
Champion 2019 [20,50]	Any school-based eHealth intervention targeting multiple health behaviours MA			Immediate Likely effective −6 trials	Ineffective −3 trials			Ineffective At both immediate and follow-up −3 trials					
				Follow-up Ineffective −3 trials									
Delgado-Noguera 2011 [46]	Any school-based nutrition intervntion MA			Free/ subsidised Ineffective −2 trials									
				Multistrategy Ineffective −7 trials									
				eHealth Likely effective −2 trials									
Evans 2012 [21,51]	Any school-based nutrition intervention MA			Likely effective −4 trials									
MacArthur 2018 [23]	Any school-based health intervention targeting multiple health behaviiours MA									Ineffective −3 trials			
Nally 2021 [28]	School-based obesity prevention interventionsMA			Ineffective −2 trials							Ineffective −4 trials		
Rose 2021 [29,47]	European school food interventions Narrative			Promising −4/6 trials		No conclusions −1 trial	Likely effective −3/3 trials				No conclusions −1 trial		No conclusions −1 trial
Singhal 2020 [30]	School-based obesity prevention interventions Narrative									Promising −9/10 trials			

Abbreviations: FV, fruit and vegetables combined; HPS, Health Promoting Schools; MA, meta-analysis; NPA, nutrition and physical activity; SSB, sugar-sweetened beverages. * Likely effective: indicating that the review found evidence of effectiveness for an intervention (if meta-analysis found an effect, or if all included primary studies were effective in narrative synthesis, as described by included review authors). Promising (more evidence needed): indicating that the review found some evidence of effectiveness for an intervention, but that more evidence is needed (if the majority (>50%) of the included primary studies in narrative synthesis were effective, as described by review authors). Ineffective: indicating that the review found evidence of lack of effectiveness for an intervention (if meta-analysis did not find an effect, or if all included primary studies in narrative synthesis were ineffective, as described by included review authors). Probably ineffective (more evidence needed): indicating that the review found evidence suggesting lack of effectiveness for an intervention, but more evidence is needed (if the majority (>50%) of included primary studies in narrative synthesis were ineffective, as described by included review authors). No conclusions possible due to lack of evidence: indicating that the review found insufficient evidence for review authors to comment on the effectiveness of an intervention (where only one primary study included in the review measured a particular dietary intake outcome, as described by included review authors). ** School interventions based on the Health Promoting Schools (HPS) framework, with active components in school education, environment, and partnerships.

**Table 3 nutrients-13-04113-t003:** Effects of RCTs of school-based nutrition interventions compared to control.

Author Year	Quality Assessment Tool	Primary Study Quality Assessment	Synthesis Method for Dietary Intake Outcomes	Summary of Findings
School-based nutrition education interventions				
Meiklejohn 2016 [44]	Unknown A validated quality criteria checklist from the American Dietetics Association Analysis Manual. Only studies with a positive or neutral rating were included	Fruit = 5 studies positive rating FV = 1 study neutral rating, 4 studies positive rating Fat = 5 studies positive rating SSB = 1 study neutral rating, 2 studies positive rating Fish = 1 study positive rating	Narrative	Fruit = 3/5 trials effective FV = 2/5 trials effective Fat = 5/5 trials effective SSB = 1/3 trials effective Fish = 1/1 trials effective NB diet in general, sucrose, water, fibre were measured, but results NR
Rahman 2017 [26]	Cochrane RoB tool	Overall, the quality of the evidence was considered moderate *	MA and narrative	SSB MA: MD −26.53 95%CI: −53.72 to 0.66, I^2^ = 6%, 2 trials, 2914 participants Narrative: 6/10 trials effective
School food environment interventions				
Bonell 2013 [45]	These criteria used for assessing methodological quality were adapted from those used in EPPI-Centre health promotion reviews	The 2 trials were strong in terms of terms of design, sample size, and adjusting for clustering in the analysis. Healthy Youth Places: “This evaluation involved a RCT design but with high attrition that differed between groups so that selection bias is a risk. The results of this study should therefore be interpreted with some caution” Middle-School Physical Activity and Nutrition: “This was a well-conducted RCT; however, the reported differences in effect were not subject to a test for interaction and so should be interpreted cautiously”	Narrative	Diet in general = 0/2 trials effective
Micha 2018 [24]	Unknown The quality of the trials was based on study design, assessment of exposure, assessment of outcome, control for confounding, and evidence of selection bias. Lower quality scores 0 to 3, higher quality 4 to 5	All 12 studies rated as higher quality	MA	Fruit = ES 0.27 95%CI: 0.09 to 0.45, I^2^ = 50%, 6 trials Vegetable = ES 0.02 95%CI −0.25 to 0.29, I^2^ = 7%, 3 trials FV = ES 0.37 95%CI: 0.05 to 0.69, I^2^ = 35%, 6 trials Fat = ES −7.15, 95%CI: −11.36 to −2.95, I^2^ = 89%, 2 trials Saturated fat = ES −2.74, 95%CI: −4.99 to −0.48, I^2^ = 90%, 2 trials SSB = ES −0.02 95%CI: −0.03 to −0.01, 1 trial Unhealthy snacks = ES −0.06, 95%CI: −0.09 to −0.02, I^2^ = 0%, 2 trials Daily caloric intake (kcal) = ES −58 95%CI: −84 to −33, 2 trials
Pineda 2021 [25,48]	Cochrane RoB2 tool	“From the RCT interventions, *n* = 38 (43%) presented a high risk of bias and *n* = 5 (12%) presented a low risk of bias” *	MA	Fruit = MD 0.2 95%CI: 0.14 to 0.26, I^2^ = 67%, 10 trials Vegetable = MD 0.00 95%CI: −0.01 to 0.01, I^2^ = 69, 7 trials
School-based nutrition interventions based on the HPS framework **				
Langford 2014 [22,49]	GRADE	Nutrition outcomes = low quality	MA	Nutrition only FV = SMD 0.15 95%: CI 0.02 to 0.29, I^2^ = 83%, 9 trials, 6210 participants Fat = SMD −0.08 95%CI: −0.21 to 0.05, I^2^ = 68%, 7 trials, 4216 participants Nutrition and PA FV = SMD 0.04 95%CI: −0.18 to 0.26, I^2^ = 79%, 4 trials, 6612 participants Fat = SMD −0.04 95%CI: −0.20 to 0.12, I^2^ = 95%, 10 trials, 12,460 participants
Other school-based nutrition interventions				
Champion 2019 [20,50]	GRADE	FV intake immediately post = moderate quality FV intake follow-up = low quality Fat consumption = low quality Unhealthy snacks (including SSB) immediately post = low quality Unhealthy snacks (including SSB) follow-up = low quality	MA	FV Immediately after intervention: SMD 0.11 95%CI: 0.03 to 0.19, I^2^ = 42%, 6 trials, 7390 participants At follow-up: SMD 0.07 95%CI: −0.01 to 0.15, I^2^ = 52%, 3 trials, 6004 participants, follow-up range: 4 months to 36 months Fat SMD −0.06 95%CI: −0.15 to 0.03, I^2^ = 52%, 3 trials, 5240 participants Unhealthy snacks (including SSB) Immediately after intervention: SMD −0.02 95%CI: −0.10 to 0.06, I^2^ = 54%, 3 trials, 5812 participants At follow-up: SMD −0.06 95%CI: −0.15 to 0.0, I^2^ = 17%, 2 trials, 2667 participants, follow-up range: 4 months to 24 months
Delgado-Noguera 2011 [46] †	Quality Assessment Tool for Quantitative Studies from the EPHPP of Ontario. Only data from studies with a strong or moderate quality were included.	eHealth = 2 studies moderate quality Free/Subsidised fruit = 2 studies moderate quality Multistrategy = 5 studies moderate quality, 2 studies strong quality	MA	FV eHealth interventions: SMD 0.33 95%CI: 0.16 to 0.50, I^2^ = 0%, 2 trials, 606 participants Free or subsidised FV distribution interventions: SMD 0.02 95%CI: −0.08 to 0.12, I^2^ = 0%, 2 trials, 1536 participants Multistrategy interventions (e.g., curriculum, environment, parent and teacher involvement): SMD 0.08 95%CI: −0.00 to 0.17, I^2^ = 50%, 7 trials, 4800 participants
Evans 2012 [21,51]	Unknown The assessment of the quality of the trials was based on the following 3 criteria: reporting of sequence generation criteria; allocation concealment; and blinding of participants, personnel, or outcome assessors.	“The quality of the 22 trials included in the meta-analyses was generally poor with evidence of high risk of bias. One study reported on all 3 criteria and was, therefore, judged to be at low risk of bias. Ten studies reported on one or 2 criteria and were, therefore, judged to be at medium risk of bias. The remaining 11 trials were judged to be at high risk of bias and did not clearly report sequence- generation criteria, allocation concealment, or blinding of participants, personnel, or outcome assessors” *	MA	FV = ES 0.26 95%CI: 0.12 to 0.40, I^2^ = 1%, 4 trials
MacArthur 2018 [23]	GRADE	Diet in general = moderate quality	MA	Diet in general = OR 0.82 95%CI: 0.64 to 1.06, I^2^ = 49%, 3 trials, 6441 participants
Nally 2021 [28]	Cochrane RoB tool	“All the studies in this review had a low risk of bias for selective reporting (*n* = 48, 100%). Approximately half of the studies were assessed as having an unclear risk of bias due to insufficient descriptions in terms of random sequence generation (*n* = 21, 44%). Most of the interventions were judged as having a low risk of bias in terms of selection bias (*n* = 38, 79%), performance bias (*n* = 37, 77%), detection bias (*n* = 34, 71%) and attrition bias (*n* = 39, 81%). The studies that were judged as having the highest risk of bias were for incomplete outcome data, blinding of outcome assessment and performance bias” *	MA	FV Portions per day = MD 0.05 95%CI: −0.08 to 0.17, 5 trials, 4741 participants Grams per day = MD 10.45 95%CI: −17.53 to 38.43, 2 trials, number of participants NR Total energy = MD 5.23 95%CI: −77.83 to 88.28, 4 trials, 1576 participants
Rose 2021 [29,47]	JBI quality assessment tool	“Overall quality scores for the 9 RCTS ranged from seven to nine out of 13. Overall, the studies were deemed to be of good quality”.	Narrative	FV = 4/6 trials effective Saturated fat = 1/1 trials effective SSB = 3/3 trials effective Total energy = 1/1 trials effective Breakfast frequency = 1/1 trials effective
Singhal 2020 [30]	Cochrane RoB tool	“Due to insufficient detail in the reporting of methods, 44.4% of judgements across all domains were ‘unclear risk of bias’. Fifteen of the 21 studies had a ‘high risk of bias’ for at least one domain (12.2% overall). Seven included studies were assessed as ‘low risk of bias’ for five or more domains. Two of the more recent trials had predominantly low risk judgements and appear to be higher quality than the rest of the field. Confidence in the results from these trials is higher than those which have likely sources of bias (one of which measured nutrition outcome)” *	Narrative	Diet in general = 9/10 trials effective

Abbreviations: CI, confidence interval; EPHPP, Effective Public Health Practice Project; EPPI, Centre Evidence for Policy and Practice Information and Co-ordinating Centre; ES, effect size; FV, fruit and vegetables combined; GRADE, Grading of Recommendations, Assessment, Development, and Evaluations; HPS, Health Promoting Schools; JBI, Joanna Briggs Institute; MA, meta-analysis; MD, mean difference; NR, not reported; OR, odds ratio, RCTs, randomised controlled trial; RoB, risk of bias; SMD, standardised mean difference; SSB, sugar-sweetened beverages. * Primary quality assessments based on all studies included in the systematic review. ** School interventions based on the Health Promoting Schools (HPS) framework, with active components in school education, environment, and partnerships. † Comparative effectiveness of alternative interventions (two trials, moderate quality): One trial found both free distribution of fruits and vegetables, and the multicomponent program were effective in increasing consumption of fruit by +0.2 portions per day. One trial reported significant increase in portions of fruits and vegetables per day in the teacher-based intervention compared to the nutritionist-based one (increase in consumption of fruit (≥2 portions per day) in 68.5% versus 48.8%, and increase in consumption of vegetables (≥2 portions per day) in 69.7% versus 31.8%), respectively.

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
