# Peer review of "School-Based Nutrition Interventions in Children Aged 6 to 18 Years: An Umbrella Review of Systematic Reviews"

_nutrients, 2021, doi:10.3390/nu13114113_

Round 1
Reviewer 1 Report
Summary:
The manuscript aims to summarize research on school-based dietary interventions among 6-18 year old children by combining available systematic reviews. The authors report findings consistent with existing research suggesting a positive impact of school-based nutrition interventions.
Comments and Suggestions:
Why were data not extracted from the primary sources?
Is it possible to elaborate more on the specific countries studied? Among "developed" countries there is a wide variety in school meal programs (cost to students, percent of students participating, etc).
Author Response
Comment 1: Why were data not extracted from the primary sources?
Response to reviewer: We followed Cochrane handbook guidelines for conducting Overviews of Reviews (also referred to as Umbrella Reviews), where “the unit of searching, inclusion and data extraction is the systematic review (and not the primary study).”
Comment 2: Is it possible to elaborate more on the specific countries studied? Among "developed" countries there is a wide variety in school meal programs (cost to students, percent of students participating, etc).
Response to reviewer: The specific countries included in the review by Meiklejohn 2016, which restricted their inclusion criteria to only developed countries, are summarised in table 1. We have also added some more detail in the manuscript text to provide some examples of the countries included in Meiklejohn 2016 (page 21)
Reviewer 2 Report
Thank you for the opportunity to review this paper on “School-based nutrition interventions in children aged 6 to 18 2 years: an umbrella review of systematic reviews”. Overall, I found this review to be informative and well conducted. I have a few small suggestions for change below:
- Please explain how 14 reviews were spread across 19 records
- Please check references. For example, I noticed that Rahman (2018; Effectiveness of Behavioral Interventions to Reduce the Intake of Sugar-Sweetened Beverages in Children and Adolescents: A Systematic Review and Meta-Analysis) is sometimes incorrectly cited as 2017 and 2021.
- In Table S3 footnote, ‘explicitly’ should be ‘explicit’.
- On page 24 at line 539 and 548, “reviews intended to synthesis intervention” should be “synthesize”
- On page 25, line 577 “two reviews assess as high quality” should be “assessed”
- Page 25, line 596 should say “generalizable to middle- and high-income countries on the”
- At line 170 and 640, it is stated that no language restrictions were imposed on the review. However, the last row of Table S1 states “limit 19 to (english language and yr="2010 -Current")”. If the review did in fact accept non-English reviews, did any make it through to the final selection? If not, is this because the other reviews imposed English restrictions?
Author Response
Comment 1: Please explain how 14 reviews were spread across 19 records
Response to reviewer: From our comprehensive search we located a number records associated with the included reviews. For example for one of the included reviews we also located a protocol paper describing the review methodology (Champion), and for another we located an abridged version of the review (Langford). These associated studies have now been cited in tables 1 to 3 alongside the relevant included reviews.
Comment 2: Please check references. For example, I noticed that Rahman (2018; Effectiveness of Behavioral Interventions to Reduce the Intake of Sugar-Sweetened Beverages in Children and Adolescents: A Systematic Review and Meta-Analysis) is sometimes incorrectly cited as 2017 and 2021.
Response to reviewer: Thank you, we have checked over the references and have amended the reference for Rahman in the supplementary file to be 2017 consistently.
Comment 3: In Table S3 footnote, ‘explicitly’ should be ‘explicit’.
Response to reviewer: Thank you, this has been amended.
Comment 4: On page 24 at line 539 and 548, “reviews intended to synthesis intervention” should be “synthesize”
Response to reviewer: Thank you, this has been amended.
Comment 5: On page 25, line 577 “two reviews assess as high quality” should be “assessed”
Response to reviewer: Thank you, this has been amended.
Comment 6: Page 25, line 596 should say “generalizable to middle- and high-income countries on the”
Response to reviewer: It is not exactly clear what the reviewer is asking, the review authors assume it was a comment referring to the use of the American spelling for ‘generalizable’ instead of the English spelling. Assuming this is correct, this has been amended in the manuscript.
Comment 7: At line 170 and 640, it is stated that no language restrictions were imposed on the review. However, the last row of Table S1 states “limit 19 to (english language and yr="2010 -Current")”. If the review did in fact accept non-English reviews, did any make it through to the final selection? If not, is this because the other reviews imposed English restrictions?
Response to reviewer: Thank you for your comment. This is an error, the search strategy was limited to English only and one review included in the umbrella review, Silveira 2011 (translated from Portuguese), has now been excluded and the manuscript has been updated accordingly (see Methods and Discussion sections). The included systematic reviews were not required to limit the primary studies by English language or geographic location.